# MsTok: Query-Based Multi-Scale 1D Visual Tokenization

## Abstract

One-dimensional (1D) image tokenizers, such as TiTok, have achieved remarkable breakthroughs in efficient image generation by encoding images into extremely compact sequences of discrete tokens. However, we identify two critical, inherent limitations in its architecture. First, TiTok's single-scale encoding strategy restricts the capability of its latent code to simultaneously capture both macroscopic structures and microscopic details of an image, creating a representation bottleneck. Second, TiTok's design, which concatenates latent tokens with image patch sequences as a unified input to the encoder, results in a quadratic increase in computational complexity when scaling the number of latent tokens, leading to an efficiency bottleneck. To address both issues concurrently, we propose MsTok, a novel, multi-scale aware, and computationally efficient 1D image tokenizer. Our approach introduces two core innovations: 1) We constructs a hierarchical multi-scale memory by aggregating selected intermediate ViT layers with scale embeddings. 2) We decouple the latent tokens from the backbone encoder and decoder, reformulating them as a set of "query tokens" that interact with the multi-scale memory through a separate and efficient cross-attention module after the image encoding is complete. This decoupled design reduces the computational cost of increasing latent tokens from a quadratic to a linear relationship. Experiments show that MsTok not only significantly improves image reconstruction quality but also demonstrates superior computational scalability to the number of latent tokens, paving the way for more powerful and efficient generative models.

## 1 Introduction

Image generation (Dhariwal & Nichol, 2021; Esser et al., 2021; Chang et al., 2022; Tian et al., 2024) has achieved notable progress, largely propelled by two distinct paradigms: diffusion models (DMs), which typically operate on continuous image representations (Rombach et al., 2022; Peebles & Xie, 2023), and autoregressive (AR) models, which model images in a discrete token space (Esser et al., 2021; Chang et al., 2022; Yu et al., 2023a; Tian et al., 2024). For the latter, the performance of the generative model is fundamentally influenced by the properties of the image tokenizer, as it translates high-dimensional pixel data into a sequence of discrete tokens. This tight coupling has motivated the community to push the boundaries of image tokenization for high-quality and efficient visual generation. Within this context, TiTok (Yu et al., 2024a) has stood out by challenging the traditional 2D grid formulation. Instead of operating on dense spatial maps, it encodes an entire image into a short 1D sequence of merely 32–64 tokens. This radical compression not only accelerates training and inference but also makes large-scale generative modeling computationally feasible.

Yet, despite its impressive performance, our analysis reveals two critical, coupled limitations in TiTok's design that curtail further progress. The first is a **representation bottleneck**: by processing visual information at a single scale, TiTok struggles to simultaneously capture both the global structure of a scene and the fine-grained details of objects within its extremely compressed token budget. This often forces a trade-off where reconstructions sacrifice texture fidelity for structural consistency, or vice versa. The second is an **scalability bottleneck**: TiTok's architecture concatenates learnable latent tokens with image patch tokens, feeding them as a unified input to its Transformer encoder. This design couples the self-attention complexity to the number of latent tokens, causing a quadratic explosion in computational cost when attempting to increase representational capacity. This creates an intractable trade-off between model expressiveness and computational scalability.

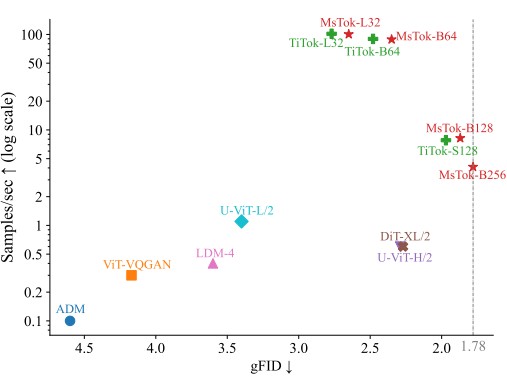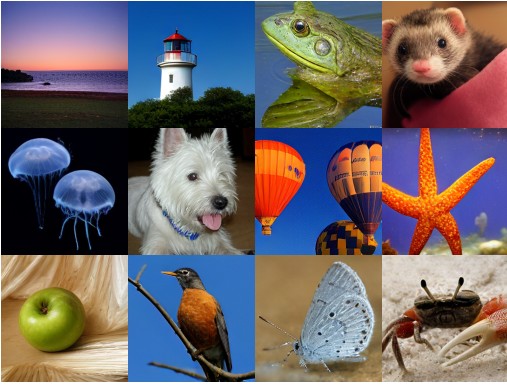

(a) Performance vs. Efficiency Comparison    (b) Generated Image Samples

Figure 1: **Overview of MsTok.** (a) Performance comparison showing MsTok achieves a better trade-off between performance (e.g., lower FID) and efficiency (e.g., higher throughput) compared to prior arts. (b) A showcase of diverse, high-fidelity images generated by our method, demonstrating its strong generative capability.

Recent studies (Yu et al., 2024b; Li et al., 2024) have highlighted the importance of high-quality visual representations in image generation. Their gains often come from introducing multi-scale features to jointly retain global layout and local texture, or leveraging stronger teacher encoders. For a ViT tokenizer, the simplest way to obtain multi-scale signals is to aggregate intermediate layers. This, however, yields a long hierarchical memory, posing a central challenge: how to compress it into a very short 1D latent sequence without discarding fine, generation-relevant cues. The query cross-attention addresses this: after the backbone finishes encoding, a small set of learnable queries selectively pools hierarchical evidence from the multi-scale memory to form compact tokens that still retain global structure and local detail. Simultaneously, this "encode–then–query" decoupling keeps the backbone cost fixed and adds only a lightweight readout whose cost grows linearly with the number of latent tokens, allowing the model to scale with minimal throughput loss.

In this work, we introduce MsTok, as shown in Figure 1, a novel 1D tokenizer that jointly addresses these two bottlenecks. Our approach first constructs a hierarchical, multi-scale feature memory by aggregating intermediate ViT layers. This explicitly provides the tokenizer with access to both high-level semantic features (from deeper layers, for global structure) and low-level fine-grained features (from shallower layers, for texture and detail), creating a richer source of information than a single-scale output. To tackle the scalability bottleneck, we decouple latent token formation from the backbone encoding process. Instead of concatenating latent tokens with image patches, a small set of learnable query tokens retrieves information from the multi-scale memory via a lightweight cross-attention module *after* the main encoding is complete. This redesign keeps the backbone complexity unchanged and turns the cost of increasing latent tokens from quadratic to linear, enabling flexible capacity scaling without prohibitive overhead. We further apply a symmetric query-based module in the decoder and adopt a two-stage training schedule that selectively optimizes the query tokens to enhance representational quality.

Our contributions are summarized as:

1. We introduce a hierarchical multi-scale feature memory built from intermediate ViT layers, which markedly improves the tokenizer's representational quality and directly translates into higher downstream generative fidelity.

2. We propose a decoupled, query-based tokenization mechanism that converts the cost of increasing latent tokens from quadratic to linear, yielding strong scalability with respect to the latent token count and enabling exploration of larger token budgets for better generators (e.g., superior gFID and throughput at numver of latent tokens=128).

3. We provide extensive experiments on ImageNet-1K. Our best model, MsTok-B-256, achieves 0.86 rFID and 1.78 gFID. Furthermore, our method exhibits better scalability with respect to the number of latent tokens.

## 2 RELATED WORKS

### 2.1 IMAGE TOKENIZATION

Image tokenizers were pioneered by VQ-VAE (Van Den Oord et al., 2017) and VQ-VAE-2 (Razavi et al., 2019), which introduced vector quantization to learn a discrete codebook for image representation with variational autoencoder (Kingma & Welling, 2013). VQ-GAN (Esser et al., 2021) enhanced reconstruction quality through adversarial training, while ViT-VQGAN (Yu et al., 2021) further incorporated Transformer architectures (Vaswani et al., 2017) into the tokenizer design. More recent variants such as Magvit (Yu et al., 2023a) achieved stronger performance by leveraging multi-task learning, RQ-VAE (Lee et al., 2022) introduced residual quantization with using multiple vector quantization steps per latent embedding, Magvit-2 (Yu et al., 2023b) and FSQ (Mentzer et al., 2023) propose a lookup-free quantization.

Although 2D tokenizers have played a foundational role in advancing high-resolution image generation and remain useful in certain scenarios, they suffer from rigidity due to the fixed spatial structure of latent grids, which limits flexibility and efficiency in representing semantic information. To overcome these limitations, TiTok (Yu et al., 2024a) introduces a 1D tokenizer that provides more compact and flexible representations, thereby achieving higher compression rates and generation efficiency. Building on this direction, SoftVQ-VAE (Chen et al., 2025) employs continuous soft quantization to avoid discretization errors while maintaining efficiency. Subsequently, One-D-Piece (Miwa et al., 2025) proposes a 1D discrete tokenizer with a tail token drop mechanism to enable quality-controllable compression. Most recently, FlexTok (Bachmann et al., 2025) extends the paradigm by introducing variable-length 1D token sequences, allowing a single model to adaptively operate across different compression rates and resolutions.

### 2.2 IMAGE GENERATION

Image generation has witnessed remarkable progress in recent years, largely driven by advances in deep generative models. Early approaches based on Generative Adversarial Networks (GANs) (Goodfellow et al., 2014) laid the foundation for learning latent representations and synthesizing realistic images. More recently, diffusion models (Ho et al., 2020) have become the dominant paradigm, leveraging iterative denoising processes to achieve state-of-the-art fidelity and diversity. In parallel, autoregressive (AR) models such as DALL·E (Ramesh et al., 2021), Parti (Yu et al., 2022), and Var-Clip (Zhang et al., 2024), which generate images token by token or scale by scale, have demonstrated strong scalability and compositionality.

Achieving high-fidelity synthesis requires latent representations that capture both high-level semantics and fine-grained details, which in turn necessitates integrating visual information across multiple scales. Early GAN-based models, such as Taming-VQGAN (Esser et al., 2021), leverage a multi-scale design by combining a convolutional VQGAN to capture local fine details with a transformer operating on compressed latent codes to model global structures, thereby enabling high-resolution and semantically coherent image generation. In diffusion-based approaches (Ho et al., 2020), the U-Net (Ronneberger et al., 2015) backbone naturally provides multi-scale pathways, where higher layers capture global semantics while lower layers refine local textures. Recently, VAR (Tian et al., 2024) adopts a multi-scale autoregressive prediction framework, progressively synthesizing images across scales to achieve scalable and high-quality generation. ImageFolder (Li et al., 2024) leverages a multi-scale folding strategy that transforms images into hierarchically aligned 1D sequences, enabling autoregressive models to capture both global structures and fine-grained details.

Beyond architectural innovations, recent studies highlight the crucial role of pretrained representations in advancing image generation. Open-MAGVIT2 (Luo et al., 2024) pretrains visual tokenizers on large-scale data via a lookup-free quantizer to achieve leading zero-shot reconstruction, and integrates these representations with autoregressive models through asymmetric token factorization for scalable visual generation. Further, REPA (Yu et al., 2024b) shows that by aligning continuous representations into a 1D sequence, diffusion transformers can be trained more easily. LlamaGen (Sun et al., 2024) demonstrates that large-scale autoregressive transformers, when equipped with strong pretrained visual tokenizers, can surpass diffusion models in both scalability and image quality.

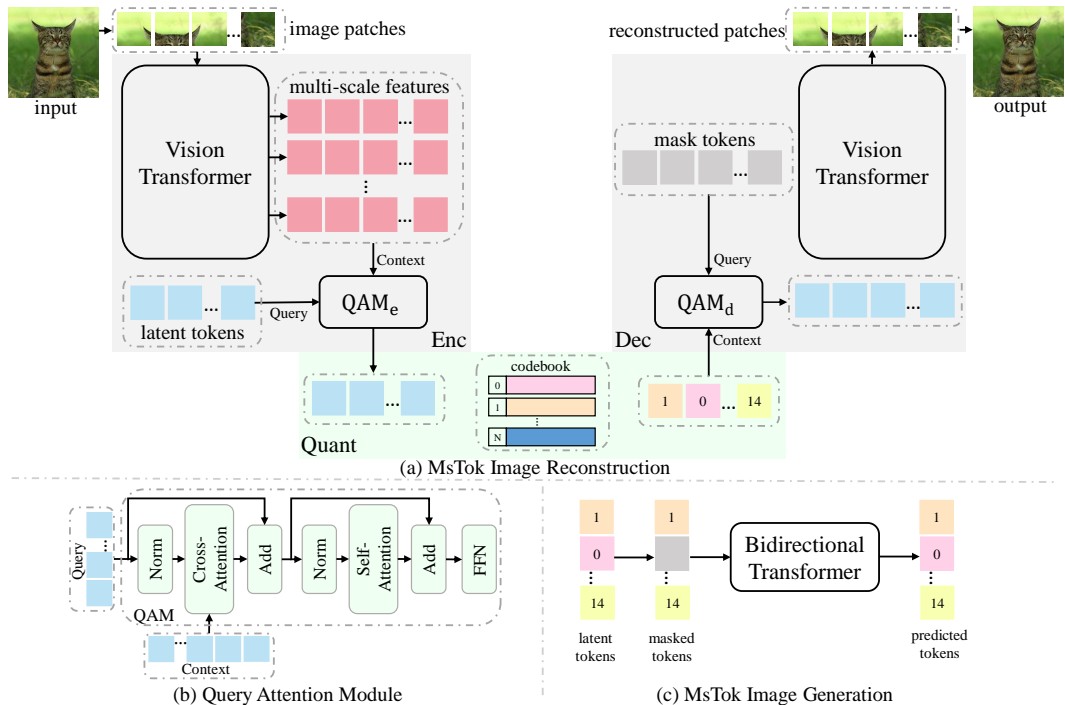

Figure 2: Illustration of (a) image reconstruction, (b) query qttention module and (c) image generation of MsTok framework (c). MsTok comprises a ViT encoder Enc that yields multi-scale features aggregated into a hierarchical memory, a query attention module $QAM_e$ where the learnable latent tokens retrieve a compact 1D latent which is vector-quantized by Quant, and a symmetric decoder Dec with $QAM_d$ that lets mask tokens attend to the quantized latent for spatial recovery.

## 3 METHOD

### 3.1 PRELIMINARY BACKGROUND

Image tokenization plays a pivotal role in modern generative models by providing a compact, discrete representation of an image in a latent space. This process has evolved from traditional 2D grid-based approaches to more recent 1D sequence-based methods, each with distinct advantages and limitations that motivate our approach.

**Traditional 2D Grid-based Tokenization.** The foundational approach, exemplified by Vector-Quantized Generative Adversarial Networks (VQ-GAN) [1], consists of three main components: an encoder Enc, a vector quantizer Quant, and a decoder Dec. Given an input image $I \in \mathbb{R}^{H \times W \times 3}$, the encoder maps it to a grid of latent embeddings $Z_{2D} \in \mathbb{R}^{\frac{H}{f} \times \frac{W}{f} \times D}$, downsampling spatial dimensions by factor $f$:

$$Z_{2D} = \text{Enc}(I). \tag{1}$$

The vector quantizer then maps each continuous embedding vector $z \in \mathbb{R}^D$ to its nearest code $c_i$ from a learnable codebook $\mathcal{C} \in \mathbb{R}^{N \times D}$:

$$\text{Quant}(z) = c_i, \quad \text{where } i = \arg\min_{j \in \{1,...,N\}} \|z - c_j\|_2 \tag{2}$$

Finally, the decoder reconstructs the image: $\hat{I} = \text{Dec}(\text{Quant}(Z_{2D}))$. However, this rigid 2D spatial correspondence limits the model's ability to exploit spatial redundancies for compact representations.

**1D Sequence-based Tokenization.** To overcome these limitations, the Transformer-based 1-Dimensional Tokenizer (TiTok) represents images as compact 1D sequences, decoupling latent size from image resolution. TiTok employs a Vision Transformer (ViT) encoder that processes flattened

image patches $P \in \mathbb{R}^{\frac{H}{f} \times \frac{W}{f} \times D}$ concatenated with $K$ learnable latent tokens $L \in \mathbb{R}^{K \times D}$:

$$Z_{1D} = \text{Enc}(P \oplus L) \tag{3}$$

where only the $K$ latent tokens are retained as the final representation. For reconstruction, a ViT decoder processes the quantized sequence $\text{Quant}(Z_{1D})$ concatenated with mask tokens $M \in \mathbb{R}^{\frac{H}{f} \times \frac{W}{f} \times D}$:

$$\hat{I} = \text{Dec}(\text{Quant}(Z_{1D}) \oplus M) \tag{4}$$

## 3.2 MULTI-SCALE 1D TOKENIZATION

While TiTok achieves remarkable compression (e.g., $K = 32$ tokens), our analysis reveals two critical limitations: (1) representation bottleneck — single-scale encoding restricts simultaneous capture of macroscopic and microscopic image features, and (2) scalability bottleneck — concatenating latent tokens with image patches creates quadratic computational complexity when scaling the number of latent tokens. To address these challenges simultaneously, we propose MsTok, as shown in Figure 2, which introduces two core innovations: multi-scale feature extraction through multiple features fusion and decoupled query-based tokenization that separates latent token processing from backbone feature extraction.

**Hierarchical Feature Extractor.** We leverages the natural hierarchical structure of Vision Transformers. Instead of treating the ViT as a monolithic encoder, we extract features from multiple layers and concatenate them. This allows us to construct a comprehensive multi-scale feature memory $\mathcal{M}$ that captures both global structures and fine details.

Given an input image $I \in \mathbb{R}^{H \times W \times 3}$, we first divide it into patches $P \in \mathbb{R}^{N_p \times D}$, where $N_p = \frac{H}{f} \times \frac{W}{f}$ is the number of patches. A ViT backbone $\text{Enc}_{\text{ViT}}$ processes these patches through $L$ transformer layers:

$$\{F_1, F_2, ..., F_L\} = \text{Enc}_{\text{ViT}}(P) \tag{5}$$

where $F_l \in \mathbb{R}^{N_p \times D}$ represents the feature map from the $l$-th layer, containing different levels of semantic abstraction.

Since ViT maintains consistent feature dimensions across all layers, we can directly extract features from multiple layers without requiring complex fusion operations. However, to ensure effective integration of features from different semantic levels, we apply normalization and scale-aware encoding before concatenation.

Specifically, we extract features from $m$ different selected scales (e.g., layers $l_1, l_2, \cdots, l_m$). For each scale $s \in \{l_1, l_2, \cdots, l_m\}$, we first apply Layer Normalization LN to stabilize the feature distributions. Then, we add learnable scale embeddings $E_s \in \mathbb{R}^{1 \times D}$ to help the model distinguish between different semantic levels:

$$\hat{F}_s = \text{LN}(F_s) + E_s \tag{6}$$

Finally, we construct our multi-scale feature memory through concatenation:

$$\mathcal{M} = \hat{F}_{l_1} \oplus \hat{F}_{l_2} \oplus \cdots \oplus \hat{F}_{l_m} \in \mathbb{R}^{m N_p \times D} \tag{7}$$

This approach elegantly exploits the uniform feature dimensionality of ViT while capturing hierarchical visual information at different semantic levels. The resulting multi-scale feature memory $\mathcal{M}$ effectively preserves both coarse structures from deeper layers and fine details from intermediate layers, enabling the subsequent tokenization process to generate more expressive latent codes.

**Decoupled Query-based Latent Representation.** In TiTok, latent tokens are concatenated with image patches before being fed to the encoder. This results in computational complexity of $O((N_p + K)^2 \cdot D)$, creating a quadratic growth bottleneck when scaling the number of latent tokens $K$.

Our method decouples the tokenization process from the backbone feature extraction. We introduce $K$ learnable query tokens $Q \in \mathbb{R}^{K \times D}$ that are independent of the ViT encoding process. These query tokens interact with the pre-computed multi-scale feature memory $\mathcal{M}$ through a cross-attention based query attention module $\text{QAM}_e$:

$$Z'_{1D} = \text{QAM}_e(\text{Query} = Q, \text{Context} = \mathcal{M}) \tag{8}$$

This operation can be intuitively understood as each query token "asking a question" to the entire feature memory and aggregating information based on relevance scores. The output $Z'_{1D} \in \mathbb{R}^{K \times D}$ is our final 1D latent representation.

**Symmetric Query-based Decoding.** To align with our encoder design and further improve reconstruction quality, we introduce the QAM module in the decoder stage as well. This design enhances decoding efficiency for both image reconstruction and generation tasks, optimizing scalability with respect to the number of latent tokens.

Specifically, after obtaining the quantized latent representation $\text{Quant}(Z'_{1D})$, we use the quantized tokens as context and learnable mask tokens $M \in \mathbb{R}^{N_p \times D}$ as queries in our $\text{QAM}_d$ module to obtain decompressed representations:

$$M' = \text{QAM}_d(\text{Query} = M, \text{Context} = \text{Quant}(Z'_{1D})) \tag{9}$$

The decompressed representations $M'$ are then directly processed by the ViT decoder for final image reconstruction:

$$\hat{I} = \text{Dec}_{\text{ViT}}(M') \tag{10}$$

This symmetric design ensures that both encoding and decoding processes maintain linear scalability with respect to the number of latent tokens, while the cross-attention mechanism allows mask tokens to dynamically attend to relevant information in the quantized representation and decompress it into full spatial representations, leading to improved reconstruction fidelity and generation efficiency compared to TiTok's direct concatenation approach.

### 3.3 TWO-STAGE TRAINING STRATEGY WITH QUERY TOKEN OPTIMIZATION

Following TiTok's approach, we recognize that training 1D VQ models presents significant challenges due to the sensitivity of the training process and the complexity of loss functions involved. The multi-scale nature of our MsTok introduces additional complexity in balancing features from different semantic levels. To address these challenges while leveraging our multi-scale architecture effectively, we adopt a two-stage training paradigm similar to TiTok.

In the first stage, we follow TiTok's proxy code strategy but adapt it for our multi-scale architecture. Instead of directly regressing RGB values with complex loss functions, we train our model using discrete codes generated by an off-the-shelf MaskGIT-VQGAN Chang et al. (2022) model as proxy targets. During this stage, the ViT encoder is trained to extract meaningful multi-scale features while the $\text{QAM}_e$ module processes these features to generate a compact 1D latent representation. The decoder, including the $\text{QAM}_d$ module for mask token processing, is trained to reconstruct proxy codes rather than RGB values. These generated proxy codes are then passed through the same off-the-shelf VQGAN decoder to produce the final RGB image outputs.

The second stage introduces our another design: while TiTok only unfreezes the decoder, we additionally unfreeze the query tokens $Q$ in the encoder's $\text{QAM}_e$ module. This selective optimization strategy allows the model to learn optimal query representations for extracting information from the multi-scale feature memory without significantly increasing computational overhead. During this stage, the learnable query tokens $Q \in \mathbb{R}^{K \times D}$ are optimized to extract the most informative and reconstruction-friendly features from the multi-scale memory $\mathcal{M}$. The decoder's $\text{QAM}_d$ module is jointly optimized to better utilize the learned query representations for high-fidelity reconstruction. The two-stage approach ensures stable training progression from proxy codes to end-to-end optimization, while the selective unfreezing prevents destabilization of pre-learned features. The optimized queries enable the decoder to receive more informative latent representations, leading to improved reconstruction fidelity compared to fixed query approaches.

## 4 EXPERIMENTS

### 4.1 EXPERIMENTAL SETUP

**Dataset.** We conduct image reconstruction and class-conditional image generation experiments on ImageNet-1K (Deng et al., 2009) at $256 \times 256$ resolution. Images are center-cropped and resized to

Table 1: Comparison of generative models on the ImageNet-1K $256 \times 256$ benchmark. All FID scores are calculated using the ADM evaluation script (Dhariwal & Nichol, 2021). †: model pretrained on OpenImages (Kuznetsova et al., 2020). ‡: model pretrained on external datasets including OpenImages and LAION (Schuhmann et al., 2022). P, S, and T denote generator parameters, inference sampling steps, and throughput (samples/sec on A100 GPU), respectively.

| tokenizer | #tokens | codebook size | rFID↓ | generator | gFID↓ | P↓ | S↓ | T↑ |
|---|---|---|---|---|---|---|---|---|
| **diffusion-based generative models** | | | | | | | | |
| Taming-VQGAN† | 1024 | 16384 | 1.14 | LDM-8 | 7.76 | 258M | 200 | – |
| VAE† | 4096×3 | – | 0.27 | LDM-4 | 3.60 | 400M | 250 | 0.4 |
| | | | | UViT-L/2 | 3.40 | 287M | 50 | 1.1 |
| VAE‡ | 1024×4 | – | 0.62 | UViT-H/2 | 2.29 | 501M | 50 | 0.6 |
| | | | | DiT-XL/2 | 2.27 | 675M | 250 | 0.6 |
| **transformer-based generative models** | | | | | | | | |
| Taming-VQGAN | 256 | 1024 | 7.94 | Taming-Transformer | 15.78 | 1.4B | 256 | 7.5 |
| RQ-VAE | 256 | 16384 | 3.20 | RQ-Transformer | 8.71 | 1.4B | 64 | 16.1 |
| | | | | | 7.55 | 3.8B | 9.7 | – |
| MaskGIT-VQGAN | 256 | 1024 | 2.28 | MaskGIT-ViT | 6.18 | 177M | 8 | 50.5 |
| ViT-VQGAN | 1024 | 8192 | 1.28 | VIM-Large | 4.17 | 1.7B | 1024 | 0.3 |
| LlamaGen | 256 | 16384 | 2.19 | LlamaGen-L | 3.80 | 343M | 256 | – |
| MAGVIT-v2 | 256 | 262144 | 1.16 | MAGVIT | 1.78 | 307M | 256 | – |
| OpenMAGVIT2 | 256 | 262144 | 1.17 | OpenMAGVIT2-B | 3.08 | 343M | 256 | – |
| One-D-Piece-B | 256 | 4096 | 1.11 | MaskGIT-ViT | 2.70 | 177M | 8 | – |
| FlexTok | 1-256 | 64000 | 1.45 | LlamaGen | 1.86 | 1.33B | 32 | 34.5 |
| TiTok-L-32 | 32 | 4096 | 2.21 | MaskGIT-ViT | 2.77 | 177M | 8 | 101.6 |
| TiTok-B-64 | 64 | 4096 | 1.70 | MaskGIT-ViT | 2.48 | 177M | 8 | 89.8 |
| TiTok-S-128 | 128 | 4096 | 1.71 | MaskGIT-UViT-L | 2.50 | 287M | 8 | 53.3 |
| | | | | | 1.97 | | 64 | 7.8 |
| MsTok-L-32 (ours) | 32 | 4096 | 2.16 | MaskGIT-ViT | 2.65 | 177M | 8 | 100.5 |
| MsTok-B-64 (ours) | 64 | 4096 | 1.65 | MaskGIT-ViT | 2.35 | 177M | 8 | 88.7 |
| MsTok-B-128 (ours) | 128 | 4096 | 1.11 | MaskGIT-UViT-L | 1.87 | 287M | 64 | 8.2 |
| MsTok-B-256 (ours) | 256 | 4096 | 0.86 | MaskGIT-UViT-L | **1.78** | 287M | 64 | 4.1 |

$256\times256$ and normalized following the preprocessing used by the proxy tokenizer (Chang et al., 2022). Results are reported on the 50k validation set.

**Tokenizer setup.** All models are trained on $256\times256$ images. We use the open-source MaskGIT-VQGAN (Chang et al., 2022) to provide discrete proxy codes during training. Following TiTok (Yu et al., 2024a), both the tokenizer and de-tokenizer adopt a patch size of $f=16$, and the proxy codebook has size $N=1024$ with code dimensionality 256. We primarily evaluate two standard model capacities MsTok-B/L with approximately 95M, and 320M parameters in the encoder and decoder, respectively. The number of query tokens $K$ is set to 64 by default, with ablation studies exploring values from 64 to 256. We extract features from 3/6, 6/12, 9/18, and 12/24-th layers of MsTok-B/L for the multi-scale features construction.

**Evaluation metrics.** We evaluate using a broad set of measures spanning both reconstruction and generation quality: reconstruction FID (rFID) and generation FID (gFID) (Heusel et al., 2017) on the ImageNet dataset. We also report training and inference throughput to enable a direct comparison of generative model efficiency as a function of latent size. In addition, because a 1D VQ tokenizer naturally serves as a compact image compressor, we assess semantic retention via linear probing under the MAE protocol (He et al., 2022). Complete details of the training and testing protocols (e.g., hyperparameters and compute) are provided in Appendix B.

**Implementation Details.** Our tokenizer training protocol largely mirrors that of TiTok (Yu et al., 2024a). We train on ImageNet for 1.5M steps using a batch size of 256 (equivalent to 300 epochs), applying standard augmentations like random cropping and horizontal flipping. The AdamW optimizer (Loshchilov & Hutter, 2017) is used with a 1e-4 learning rate and 1e-4 weight decay. The learning rate schedule includes a warm-up period followed by cosine decay, and we clip gradients at a threshold of 1.0. To stabilize training and enhance quality, we employ several techniques: an Exponential Moving Average (EMA) with 0.999 decay is applied to the model weights, and all reported results are from the EMA model. We also incorporate a discriminator loss (Esser et al., 2021) and utilize proxy codes from MaskGIT (Chang et al., 2022). During the final 500k steps, we

freeze the encoder and train only the decoder. For the downstream generation task, the generator's configuration is aligned with MaskGIT. It is trained for 500k steps with a batch size of 2048, using the AdamW optimizer and a constant 1e-4 learning rate. Random horizontal flipping is the only data augmentation used. An EMA with a 0.999 decay is also applied. During inference, we use either 8 or 64 sampling steps and employ classifier-free guidance to boost image quality.

## 4.2 MAIN RESULTS

**Tokenizer Comparison.** We first assess the multi-scale representational quality of MsTok on the image reconstruction task. MsTok is compared against the original TiTok (used as our baseline) and several state-of-the-art tokenizers. We report the key rFID of each method, as summarized in Table 1. Compared with baseline TiTok, introducing multi-scale features leads to superior reconstruction fidelity. In particular, MsTok-B-64 and MsTok-L-32 obtain rFID of 1.65 and 2.16, respectively. Because MsTok decouples latent tokens from feature extraction, it scales more gracefully with the number of tokens. Accordingly, we further evaluate lengths 128 and 256: MsTok-B-128 and MsTok-B-256 reach rFID 1.11 and 0.86 with substantially smaller efficiency degradation than TiTok. Figure 3 further demonstrates the efficiency advantage of our method over TiTok during tokenizer pretraining. Compared to TiTok, whose training time grows rapidly as the latent length $K$ increases, MsTok maintains nearly constant pre-training throughput across a wide range of $K$, demonstrating strong scalability with respect to the latent token length.

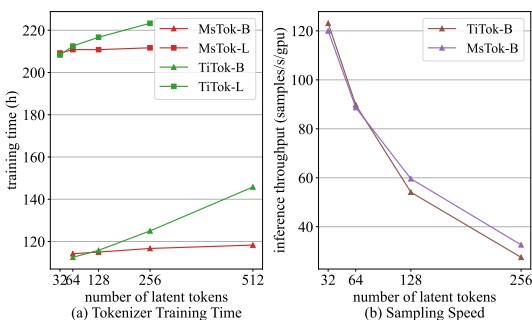

Figure 3: Scalability of MsTok. (a) Stage-2 training time vs. latent token count $K$ for TiTok and MsTok. (b) Sampling speed (images/s) of TiTok and MsTok at generation time under different $K$.

**Class-conditional image generation.** The quality of the learned discrete tokens directly impacts the performance of downstream generative models. We evaluate this by training a class-conditional generator on the tokens produced by MsTok, with results summarized in Table 1. Compared to the baseline TiTok, our method demonstrates superior generation quality and efficiency. Specifically, MsTok-L-32 and MsTok-B-64 achieve gFID scores of 2.65 and 2.35, respectively, which represent improvements of 0.12 and 0.13 over their TiTok counterparts while maintaining a considerable lead in throughput. Furthermore, thanks to our design that decouples latent tokens from the feature extraction of backbone, MsTok exhibits superior scalability when increasing the number of latent tokens. For instance, our MsTok-B-128 not only achieves a better gFID than TiTok-S-128 (1.87 vs. 1.97) but also boasts a higher throughput (8.2 vs. 7.8 samples/sec), fully validating the excellent scalability of our approach. By further increasing the token count, MsTok-B-256 achieves a state-of-the-art gFID of 1.78, highlighting the effectiveness and scalability of our method.

Table 2: Linear probing results on ImageNet-1K. We report top-1 classification accuracy by training a linear classifier on frozen latent tokens from different tokenizers.

| Method | L-32 | B-64 | B-128 | B-256 |
|--------|------|------|-------|-------|
| TiTok | 60.0 | 53.9 | 53.6 | 52.1 |
| MsTok | **61.7** | **57.3** | **56.4** | **54** |

**Representation Quality.** A high-quality tokenizer should not only enable faithful reconstruction but also produce latent tokens that retain the core semantic content of the image. To assess this "semantic retention," we employ linear probing, a standard method for evaluating the quality of learned representations. Following the protocol from MAE He et al. (2022), we freeze our pre-trained MsTok encoder and use its output latent tokens as fixed feature representations for each image in the ImageNet dataset. A single linear classification layer is then trained on top of these frozen tokens to predict the image class. As shown in Table 2, MsTok achieves significantly higher top-1 classification accuracy compared to the TiTok baseline. This result strongly indicates that our latent tokens are not just compressed data points but are semantically meaningful representations. The superior performance can be attributed to our core design: the multi-scale feature memory

Table 3: Ablation studies on the ImageNet-1K benchmark. We analyze the impact of MsTok's core components, the number of feature scales, and different module designs for feature aggregation and querying. The final settings are labeled in gray.

(a) MsTok configuration. Results reported in an cumulative manner.

| Method | rFID↓ | IS↑ |
|---|---|---|
| Baseline (TiTok) | 1.70 | 194.0 |
| + Decoupled Querying | 1.73 | 196.7 |
| + Unfreeze Query Tokens | 1.71 | 197.7 |
| + Multi-Scale Features | 1.66 | 201.8 |
| + Scale Embeddings | 1.65 | 202.9 |

(b) Effect of number of feature scales used for memory.

| Feature Levels | rFID↓ | IS↑ |
|---|---|---|
| 1 | 1.71 | 197.7 |
| 2 | 1.68 | 199.0 |
| 3 | 1.65 | 202.9 |
| 4 | 1.65 | 203.3 |

(c) Feature aggregation and querying module selection.

| Module Configuration | rFID↓ | IS↑ |
|---|---|---|
| WeightedAdd + Linear | 1.80 | 187.1 |
| Concat + Linear | 1.78 | 185.2 |
| WeightedAdd + XAttn | 1.68 | 198.6 |
| Concat + XAttn | 1.65 | 202.9 |

allows the model to capture a richer hierarchy of visual concepts, while the decoupled querying mechanism enables the model to distill this information more effectively into the final compact tokens. This demonstrates that MsTok produces a more structured and semantically aware latent space, making it highly suitable for downstream understanding tasks in addition to generation.

## 4.3 ABLATION STUDIES

We conduct comprehensive ablation studies to validate our design choices in Table 3.

**Impact of Core Components.** Table 3a shows the cumulative impact of each core component of MsTok. Starting from the TiTok baseline (rFID 1.70), we incrementally add our proposed modules. Decoupling the querying mechanism and unfreezing the query tokens slightly degrades the rFID to 1.71, suggesting that without access to richer features, the decoupled design alone is insufficient. However, the introduction of multi-scale features significantly improves performance, lowering the rFID to 1.66. Finally, adding learnable scale embeddings, which help the model distinguish between hierarchical features, provides the last boost, reaching our final rFID of 1.65. This step-by-step analysis demonstrates that our performance gains are a result of the synergistic combination of all proposed components, with multi-scale feature integration being the most critical factor.

**Effect of Multi-Scale Feature Levels.** We investigate how the number of feature levels used to construct the memory affects reconstruction quality in Table 3b. A single-scale model (equivalent to a decoupled TiTok) yields a rFID of 1.71. Performance steadily improves as we increase the number of feature levels to 2 (rFID 1.68) and 3 (rFID 1.65). Increasing the number of levels to 4 results in similar performance, indicating that the benefits begin to saturate. However, since the Inception Score (IS) continues to improve, we select 4 levels as our default configuration.

**Feature Aggregation and Querying Design.** Table 3c validates our architectural choices for feature aggregation and querying. We compare two aggregation methods (WeightedAdd vs. Concat) and two querying mechanisms (Linear vs. Cross-Attention). The results show that the querying mechanism is the dominant factor. Models using a simple linear projection perform poorly (rFID 1.80 and 1.78), regardless of the aggregation method. In contrast, employing cross-attention (XAttn) for querying yields a substantial performance boost (rFID 1.68 and 1.65). Between the aggregation methods, simple concatenation slightly outperforms weighted addition when paired with cross-attention. Our final design, combining feature concatenation with cross-attention, achieves the best performance (rFID 1.65), confirming the effectiveness of our chosen modules.

## 5 CONCLUSION

We presented MsTok, a 1D discrete image tokenizer that couples a hierarchical multi-scale feature memory with a decoupled query-based latent formation mechanism. By aggregating intermediate ViT layers into a scale-aware memory and deferring latent extraction to a lightweight cross-attention readout, MsTok alleviates the representation bottleneck of single-scale designs while converting latent scaling cost from quadratic to linear. A symmetric query module in decoding and a two-stage schedule with selective query optimization further improve reconstruction fidelity and semantic retention. Experiments on ImageNet-1K show consistent gains in rFID, gFID, linear probing accuracy, and demonstrate graceful scaling as latent length increases.

**Reproducibility Statement.** The core MsTok source code is included in the supplementary materials. All hyper-parameters for training tokenizer and generator are specified in the main paper and Appendix B. Together these resources enable faithful reproduction of our results.

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

# APPENDIX

## A THE USE OF LARGE LANGUAGE MODELS(LLMS)

We acknowledge the use of large language models (LLMs) as writing assistants only for grammaticaand style. LLMs are not employed in the core research methodology, experimental design, dataanalysis, or generation of research findings presented in this paper. All textual content has beerrigorously reviewed and verified by the authors to ensure accuracy and authenticity of the researchcontributions.

## B EXPERIMENTAL DETAILS

**Tokenizer and Generator Training.** In our experiments, we mostly follow the TiTok settings (Yu et al., 2024a) as detailed in Table 4 for both model architecture and training configurations: All models are trained on ImageNet-1K at $256 \times 256$. We adopt MaskGIT-VQGAN (Chang et al., 2022) proxy codes (codebook size $N$=1024, code dim 256) following TiTok (Yu et al., 2024a). Tokenizer and de-tokenizer use patch size $f$=16. We evaluate two capacities (MsTok-B / MsTok-L) with roughly 95M / 320M encoder+decoder parameters. The number of latent tokens is $K \in \{64, 128, 256\}$. For multi-scale memory, we extract normalized intermediate layer features from layers $\{3,6,9,12\}$ (B) and $\{6,12,18,24\}$ (L), forming 4 scales with learnable scale embeddings. Tokenizer training runs 1.5M steps (300 epochs) with batch size 256, AdamW (lr=1e−4, weight decay=1e−4), linear warm-up then cosine decay, gradient clipping at 1.0, and EMA (decay 0.999) whose weights are used for reporting. Standard data augmentation: random resized crop + horizontal flip. A discriminator loss (Esser et al., 2021) is included. The first 1M steps follow the stage 1 training. During the final 500k steps, the encoder is frozen and only the decoder and latent tokens are updated. For downstream class-conditional generation we use the MaskGIT generator configuration: 500k steps, batch size 2048, AdamW with constant lr=1e−4, horizontal flip only, EMA 0.999. Inference uses 8 or 64 sampling steps with classifier-free guidance.

**Linear Probing Hyperparameters.** In our experiments, we follow the MAE settings (He et al., 2022) as detailed in Table 4: For linear probing we freeze the entire tokenizer (ViT encoder + multi-scale memory + query cross-attention) and extract the continuous query outputs *before* vector quantization (default $K$={32, 64, 128, 256}). Unless stated, we mean-pool the $K$ latent vectors after LayerNorm to obtain a single $1 \times D$ representation. A linear classifier is trained upon the representation on ImageNet-1K train split for 90 epochs using LARS optimizer, with 10 warmup epochs, batch size 16384 and label smoothing 0, weight decay 0. Initial learning rate follows the MAE scaling rule: lr $= 0.1 \times$ batch/256. We use cosine decay for lr schedule. Weight decay $1 \times 10^{-4}$, no gradient clipping. A batchNorm layer is added before the linear head. We allpy simple RandomResizedCrop as the data augmentation.

Table 4: Hyperparameters for tokenizer MsTok (left) and generator MaskGIT (right). These hyper-parameters almost follow TiTok settings.

(a) MsTok hyperparameters.

| Item | Value |
|---|---|
| **Model** | |
| Codebook Size | 4,096 |
| Token Size | 12 |
| Model Size | ViT base / large |
| Patch Size | 16 |
| Latent Tokens | 64,128,256 / 32 |
| Layer Selection | $\{3,6,9,12\}/\{6,12,18,24\}$ |
| Feature Levels | 4 |
| **Training** | |
| Stage1 Epochs | 200 |
| Stage2 Epochs | 100 |
| Stage1 Batch Size | 256 |
| Stage2 Batch Size | 256 |
| Dataset | ImageNet-1K |
| Augmentation | Rand. Crop / Flip |
| **Losses** | |
| Stage1 Pretrained | MaskGIT tokenizer |
| Target Codebook Size | 1024 |
| Recon Weight | 1.0 |
| Quantizer Weight | 1.0 |
| Stage2 Disc. Weight | 0.01 |
| Perceptual Model | ConvNeXT-S |
| Perceptual Weight | 0.1 |
| Commit Weight | 0.25 |
| Codebook Loss W. | 1.0 |
| **Optimizer** | |
| Optimizer | AdamW |
| LR | $1 \times 10^{-4}$ |
| Beta1 / Beta2 | 0.9 / 0.99 |
| Weight Decay | $1 \times 10^{-4}$ |
| Epsilon | $1 \times 10^{-8}$ |
| **Scheduler** | |
| Type | Cosine |
| Warmup Steps | 10,000 |
| End LR | $1 \times 10^{-5}$ |

(b) MaskGIT hyperparameters.

| Item | Value |
|---|---|
| **Model** | |
| Architecture | MaskGIT-ViT / UViT-L |
| Hidden Dim | 768 / 1024 |
| Hidden Layers | 24 / 20 |
| Attn Heads | 16 / 16 |
| Dropout | 0.1 |
| Class Label Drop | 0.1 |
| Class Count | 1000 |
| Latent Tokens | 256 |
| **Training** | |
| Epochs | 800 |
| Batch Size | 2048 |
| Dataset | ImageNet-1K |
| Augmentation | Random Flip |
| **Losses** | |
| Loss | CrossEntropy |
| Label Smoothing | 0.1 |
| Unmasked Token Loss | 0.1 |
| **Optimizer** | |
| Optimizer | AdamW |
| Learning Rate | $2 \times 10^{-4}$ |
| Beta1 / Beta2 | 0.9 / 0.96 |
| Weight Decay | 0.03 |
| **Scheduler** | |
| Type | Cosine |
| Warmup Steps | 10,000 |
| End LR | $1 \times 10^{-5}$ |

## C    VISUALIZATION

We provide the visulization of generated images from our MsTok with MaskGIT (Chang et al., 2022) across random ImageNet classes in Figure 5 and with some specific ImageNet class.

## D    LIMITATIONS

Our work now rely on fixed layer selection and manual token count. Our future work may explore adaptive scale weighting, dynamic latent allocation, and extension to video or multimodal genera-tion.

Table 5: Hyperparameters for linear probing evaluation. These hyperparameters fully follow MAE settings.

| Item | Value |
|---|---|
| optimizer | LARS (You et al., 2017) |
| base learning rate | 0.1 |
| weight decay | 0 |
| optimizer momentum | 0.9 |
| batch size | 16384 |
| learning rate schedule | cosine decay |
| warmup epochs | 10 |
| training epochs | 90 |
| augmentation | RandomResizedCrop |

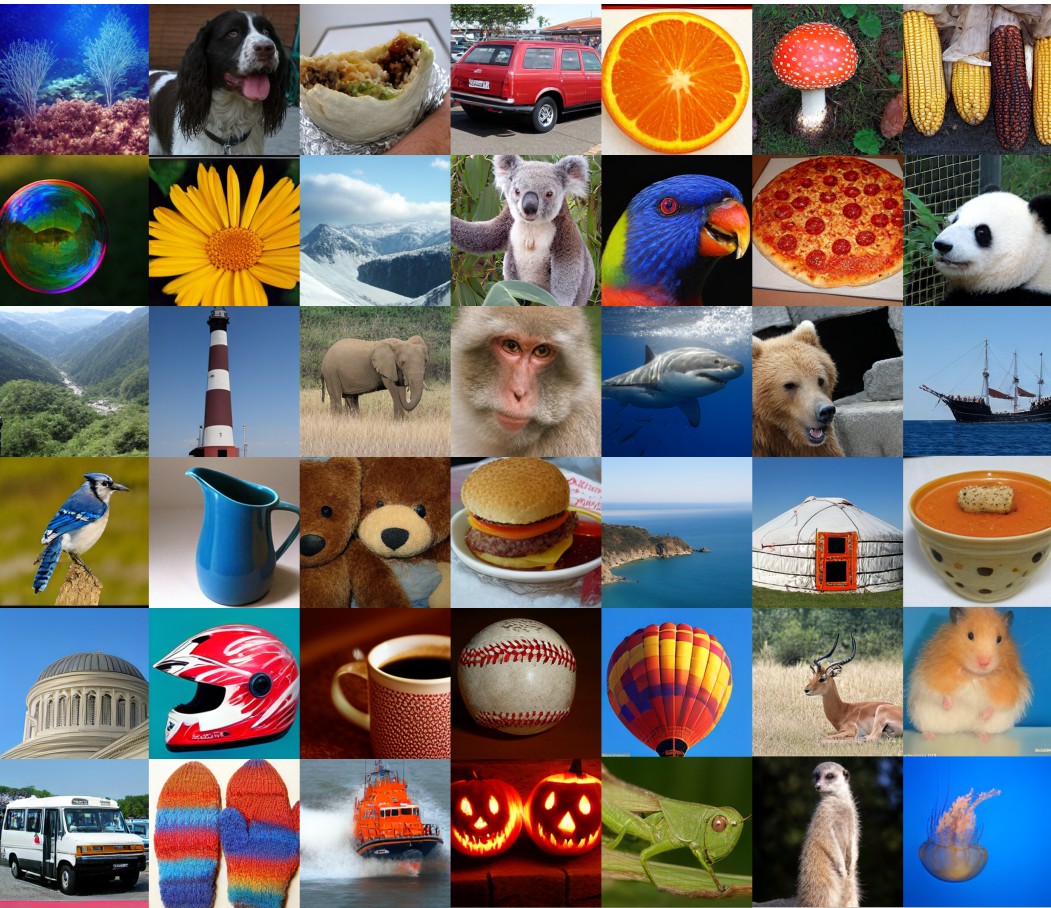

Figure 4: Visualization of generated images from MsTok-B-256 with MaskGIT (Chang et al., 2022) across random ImageNet classes.

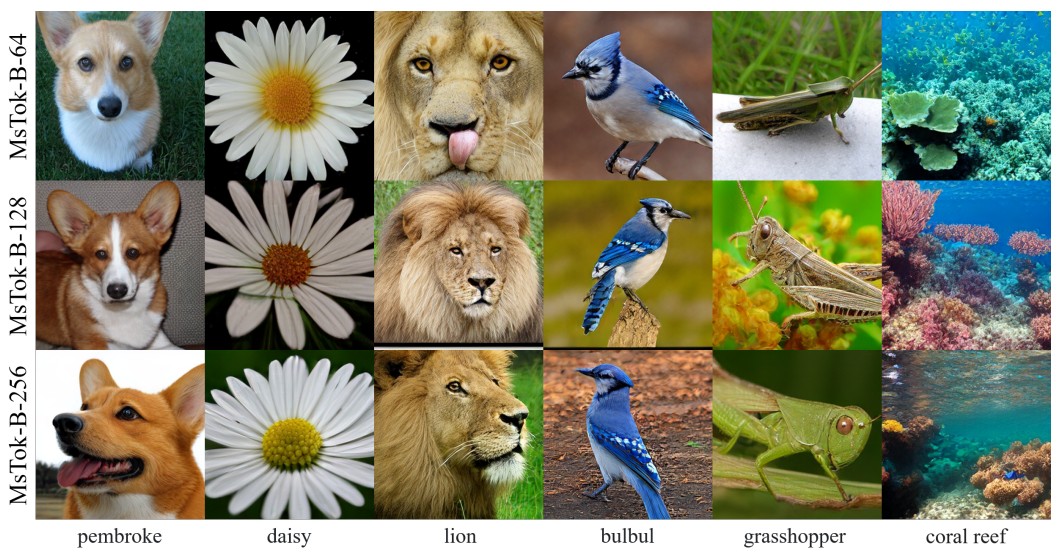

Figure 5: Visualization of generated images from MsTok-B-64, MsTok-B-128 and MsTok-B-256 with MaskGIT (Chang et al., 2022) with specific ImageNet classes.

