# OpenReview forum: "MsTok: Query-Based Multi-Scale 1D Visual Tokenization"
_ICLR.cc/2026/Conference — ICLR 2026 Conference Withdrawn Submission_

### Official Review · Reviewer_urSe · 2025-10-15

**Soundness:** 3
**Presentation:** 2
**Contribution:** 2
**Rating:** 4
**Confidence:** 4

**Summary:**

MsTok is a "multi-scale" query-based variant of TiTok that replaces TiTok’s joint self-attention encoder with a cross-attention readout from a pre-computed ViT feature bank. It keeps much of the same mechanisms intact, including vector quantization, proxy-code supervision, and MaskGIT-style decoder. Architecturally, it is closer to a Perceiver IO or latent reader than to a standard ViT.

The central idea behind "1D tokenization" in TiTok [1] is clever; it looks to encode a set of latent tokens (not a "sequence") that can be conditioned for direct reconstruction of an image (whether this represents "1D" or "tokenization" is debatable). The training is done by an initial training step where one matches an existing codebook from a pre-trained model, so the model is weakly supervised by an existing model. The original implementation uses self-attention with concatenated patches and latents.

This reviewer reads two central contributions to improve on the original; (2) the cross-attention (CA) replacement in QAM lowers complexity and enables higher $K$ latents, while (2) the "multi-scale" addendum looks to exploit several depths (not scales) from the ViT computational pipeline to provide more granular features to the latents. Both (1) and (2) are applied directly to TiTok to improve generative performance.

The CA module (1) has been thoroughly explored in other works, such as Perceiver [2,3], CrossMAE [4], and CAPI [5], to name a few sources. It is hence not an entirely novel idea, but an effective use of a well known architectural modification.

With regards to "multi-scale" features (2), the central idea is to extract features from different depths or levels of the model. This was proposed in Segmenter [6] and has been adopted in several other works. Unfortunately, the "multi-scale" term is a bit of a misnomer, and these represent a multi-level feature integration more than multiple "scales" in the classic pyramid / CNN sense.

Given that these two are well known mechanisms, the paper is somewhat incremental in nature, but has some interesting ideas and demonstrate empirical results that could very well be beneficial to the research community.

---

[1] [An Image is Worth 32 Tokens for Reconstruction and Generation, Yu et al. 2024](https://arxiv.org/abs/2406.07550)

[2] [Perceiver: General Perception with Iterative Attention, Jaegle et al. 2021.](https://arxiv.org/abs/2103.03206)

[3] [Perceiver IO: A General Architecture for Structured Inputs & Outputs, Jaegle et al. 2021.](https://arxiv.org/abs/2107.14795)

[4] [Rethinking Patch Dependence for Masked Autoencoders, Fu et al. 2024](https://arxiv.org/abs/2401.14391)

[5] [Cluster and Predict Latent Patches for Improved Masked Image Modeling, Darcet et al. 2025](https://arxiv.org/abs/2502.08769)

[6] [Segmenter: Transformer for Semantic Segmentation, Strudel et al. 2021](https://arxiv.org/abs/2105.05633)

**Strengths:**

1. The cross-attention replacement demonstrates clear computational benefits, and are built on established existing methods.
2. The multi-level approach improves reconstruction performance, as well as linear probing results.
3. These two ideas have a degree of synergy, and appear as reasonable and intuitive fixes to the original baseline. While it seems a very natural continuation, the results show clear improvements. Ablations are designed well to support this synergy, showing that the effects solve the two issues with the original work in conjunction.
4. The authors outline training setups and hyperparameters in detail, which helps improve reproducibility and transparency.

Overall, the work is intuitive and reasonable, fusing two ideas to solve some existing weaknesses in the original approach.

**Weaknesses:**

1. On the whole, the contributions are incremental. This isn't necessarily detrimental in and of itself, but the authors choose not to cite influential papers where some of these ideas have been originally explored. This would position the paper more clearly in the literature, and clarify the actual novelty of the contributions.
2. Tying the title of the paper to "multi-scale" is unfortunate. There is arguably no actual variability of classical scale involved in the features, but instead, a fusion of semantic levels from within the model. While this may very well be a conscious choice to delineate the current work from the "1D sequence" in TiTok (which again represents unfortunate phrasing), the terminology is simply not accurate. See below for further discussion.
3. The presentation of the paper is a little off. Figure 2 seems to obfuscate rather than clarify the central ideas in the paper. Some phrasing is unclear, and articles are dropped ("Since ViT maintains" [L245]). This is a minor weakness, but should be corrected in a potential camera ready version. Additionally, there is some unclear results in the ablations, see questions below.
4. As inherited from TiTok, the proposed method is wholly dependant on a pre-trained model for training. While the authors do not set out to improve on this, it is an inherent limitation of the approach, even if inherited by the original model.
5. Limitations (Appendix D, L594) is not particularly forthcoming, and seems like an afterthought. The purpose of limitations is to demonstrate genuine reflection and consideration of current limitations. As it reads in the current draft, it does not come across as reflective by the authors.

Continuing from point 2:

> Since ViT maintains consistent feature dimensions across all layers, we can directly extract features from multiple layers without requiring complex fusion operations.

Here, the authors clearly emphasize that there is no "multi-scale" involved in a classic ViT. Hence, the choice to *posit the paper on the architecture being "multi-scale"* comes across as somewhat dubious.

In summary, the major issues this reviewer has with the paper is related to implicit claims of "multi-scale" features, the incremental nature of the contributions, lack of citations and minimal disclosure of limitations for the work.

**Questions:**

1. Table 3: Why is inception score higher in Table 3b than in Tables 3a, 3c? Is there a particular reason why the exact same FID yields lower IS for this particular ablation?
2. In what sense do the authors claim the features are multi-scale, or hierarchical?
3. Do the authors have any ideas for solving the explicit dependency on pretrained models to train TiTok or MsTok?
4. MAE style pre-training is in service of an instance based classification objective. The authors show that MsTok performs adequately in this regard as well, which is interesting. Have the authors considered fully training a model with this in mind, beyond linear probing?
5. In Table 2; why does performance drop when $K$ increases?

---

### Official Review · Reviewer_GMmP · 2025-10-22

**Soundness:** 3
**Presentation:** 3
**Contribution:** 2
**Rating:** 2
**Confidence:** 3

**Summary:**

This paper proposes improvements to the 1D tokenization architecture introduced by TiTok. In particular, self-attention between 1D latent tokens and 2D visual tokens is replaced with cross-attention in both encoder and decoder, and the cross-attention encoder is tweaked to utilize context features aggregated from multiple ViT layers. These proposed changes lead to improvements in runtime that are more significant when scaling to larger numbers of latent tokens, as well as improvements in reconstruction and generation quality.

**Strengths:**

The architectural improvements are sound and natural extensions of TiTok. The use of features from several transformer layers is demonstrated to be effective, and the supporting experimental evaluation is convincing.

**Weaknesses:**

In the case of 32 or 64 tokens, there is no improvement to runtime/throughput, with only a marginal improvement to reconstruction and generation quality (around 0.1 improvement in FID). While this demonstrates that the proposed changes are sensible, it positions the contribution as a relatively minor, incremental improvement over the existing TiTok.

Furthermore, the main benefit of 1D tokenization is the fact that using spatially-decoupled tokens with the ability to represent global, rather than spatially localized attributes of the image, enables scaling of the compression ratio significantly beyond what was previously demonstrated with more traditional “2D” tokenizers. In particular, TiTok’s 32-token model offers a much lower number of tokens compared to standard 16x16 token grids produced by the standard VQGAN approach, leading to orders of magnitude improvements in generation efficiency and tokens which are very semantically rich. Therefore, the motivation for enabling scalability to larger numbers of latent tokens — which is one of the core aspects of this paper — is not very strong. In fact, the 256-token MsTok model does not appear to offer any benefits over 2D tokenizers such as MaskGIT’s VQGAN in terms of compression ratio, and achieves only marginally better reconstruction and generation than more recent 2D tokenizers such as MAGVITv2. A discussion and analysis of why 1D tokenization could enable a higher performance ceiling than 2D tokenization, even when increasing the number of tokens to match that of standard 2D tokenizers, would be a valuable addition to this paper. Without it, the motivation for using 1D tokens in the high-token-count case is weak.

Finally, despite the suggested improvements, the training strategy of MsTok remains identical or very similar to TiTok’s multi-stage training. In particular, it is necessary to use “proxy codes” in a first stage, followed by RGB fine tuning with perceptual and adversarial loss. This is a weakness of TiTok and it remains unaddressed.

Overall, despite sound methodology and experiments demonstrating great performance, my score is not higher because the proposed tweaks to TiTok are relatively straightforward architectural changes and the improvements in performance, while present, are small. I believe that further motivation and experiments to support either (a) the significance of scaling 1D tokenizers to large numbers of tokens or (b) showcasing additional benefits (beyond marginal performance improvements) that your proposed changes enable, would be important additions to this paper.

**Questions:**

**Q1.** Do you always train your encoder model from scratch (following TiTok’s two-stage training protocol with proxy codes at first)? Would it make sense to consider using large-scale pretrained ViT encoder models (e.g. DINOv2, CLIP, etc.)? Your design seems very well suited to leverage a pretrained encoder, and it would be a valuable observation if you can demonstrate that using such a large-scale pretrained encoder can lead to one or more of…
* faster training,
* a simplified training recipe,
* successful application to larger, more diverse datasets than ImageNet (LAION, OpenImages),
* improved performance.

**Q2.** Have you performed any experiments to determine the receptive field of each token? I wonder if your large token count models learn any “global” features, or whether they are mostly spatially localized, similar to a standard 2D tokenizer.

**Q3.** Could your proposed multi-scale feature extraction enable scaling of 1D tokenizers to lower numbers of tokens, e.g. 24, 16, or even less? A smaller number of tokens could unlock even faster generation speeds compared to other 1D tokenizers and potentially produce even more semantically rich representations.

---

Minor comments (typos/writing):
* L270: explanation of cross attention as “query asking a question” sounds a bit LLM-generated. In this case, it is OK to assume familiarity of the reader with the (cross) attention mechanism, so this type of high level analogy feels out of place and can be skipped.
* Overall, some additional proofreading would be good. E.g. “numver” -> “number”, “we constructs” -> “we construct”, “we leverages” -> “we leverage”, “captures global structures” -> “captures global structure”, several missing spaces in Appendix A, “Related works” -> “Related work”, etc.

---

### Official Review · Reviewer_9p5J · 2025-10-30

**Soundness:** 3
**Presentation:** 3
**Contribution:** 3
**Rating:** 4
**Confidence:** 3

**Summary:**

This paper proposes MsTok, a query-based, multi-scale 1D visual tokenizer designed to overcome the limitations of existing 1D tokenizers such as TiTok. MsTok introduces two key innovations: Hierarchical Multi-Scale Memory and Decoupled Query-Based Tokenization. Experiments on ImageNet-1K (256×256) show that MsTok achieves state-of-the-art performance among 1D tokenizers, outperforming TiTok while maintaining higher throughput and scalability

**Strengths:**

1. The paper identifies concrete architectural limitations in TiTok and systematically addresses both representational and computational challenges through a clean, modular design.
2. The experimental results are thorough, including reconstruction (rFID), generation (gFID), linear probing for classification, and ablations over feature scales, aggregation strategies, and query mechanisms (Table 3a–3c).

**Weaknesses:**

1. Although introducing additional computational overhead through multiscale design, the results in Table 3a/3b demonstrate that marginal performance gains have been observed.
2. The training protocol and generator largely rely on MaskGIT, it would be better to include autoregressive image generation results.
3. All results are confined to ImageNet-1K at 256×256. Broader validation on higher resolutions (e.g., 512×512) or different domains (e.g., COCO, FFHQ) would strengthen generality claims.

**Questions:**

Please refer to the weakness part.

---

### Official Review · Reviewer_1Nxm · 2025-11-01

**Soundness:** 3
**Presentation:** 3
**Contribution:** 3
**Rating:** 6
**Confidence:** 3

**Summary:**

This paper addresses two key limitations of existing 1D image Tokenizers (e.g., TiTok): the representation bottleneck (single-scale encoding struggles to simultaneously capture macroscopic structures and microscopic details) and the efficiency bottleneck (the concatenation of latent tokens with image patches leads to quadratic growth in computational complexity as the number of tokens increases). To tackle these issues, it proposes MsTok, a query-based multi-scale 1D visual Tokenizer. Its core designs include: 1) Constructing a hierarchical multi-scale feature memory by aggregating intermediate ViT layer features and incorporating scale embeddings; 2) Decoupling latent tokens from the backbone network, extracting information from the multi-scale memory as "query tokens" via a cross-attention module—this ensures computational complexity grows linearly with the number of tokens; 3) Adopting a symmetric query-based decoding module and a two-stage training strategy (optimizing query tokens to enhance representation quality). Experiments demonstrate that MsTok achieves superior image reconstruction (minimum rFID of 0.86) and generation performance (minimum gFID of 1.78) on the ImageNet-1K dataset. It also exhibits better computational scalability as the number of latent tokens increases, while linear probing tasks verify the semantic retention capability of its latent representations.

**Strengths:**

1. Quality and Reliability: The experimental design is rigorous. The ablation experiments (Table 3) verify the necessity of core components (decoupled querying, multi-scale features, and scale embeddings) one by one. The linear probing experiments (Table 2) supplement evidence for the semantic retention capability. The results show strong consistency with no obvious data contradictions.
2. Clarity of Description: In the method section, formulas (5)-(10) clearly define the mathematical logic of hierarchical multi-scale memory construction, query interaction, and decoding. Figure 2 intuitively demonstrates the "encoding-query-decoding" process, which lowers the barrier to understanding the complex design.

**Weaknesses:**

1. Insufficient Breakthrough in Core Innovations: The design of "constructing a hierarchical multi-scale memory bank" is simplistic and lacks sufficient innovation. This scheme essentially consists of "ViT intermediate layer feature concatenation + scale embeddings", while "multi-layer feature fusion" has been widely applied in works such as VQ-GAN and DiT. The paper fails to propose a new feature fusion mechanism, making it difficult to qualify as a "genuine innovation".
2. Lack of Details in Multi-Scale Design: The paper does not explain "the basis for selecting intermediate layers"—for example, why earlier/later layers are not chosen, or how the information complementarity of features from different layers is verified. It also does not analyze "the functional boundaries of scale embeddings", such as the specific reasons for performance degradation when scale embeddings are removed.
3. Insufficient Connection to Cutting-Edge Multimodal Scenarios: The paper only verifies unimodal image reconstruction and generation tasks on ImageNet-1K, and does not explore the performance of MsTok in multimodal tasks (e.g., image-text alignment, cross-modal generation). However, one of the core values of 1D Tokenizers is adapting to sequence-modeling MLLMs. The lack of multimodal validation weakens the breadth of its application scenarios.

**Questions:**

1. Regarding the design of the multi-scale memory bank: The paper selects specific intermediate layers of ViT (e.g., layers 3/6/9/12 for MsTok-B) to construct the memory bank. What is the basis for selecting these layers? Has there been an experimental comparison of the impact of "selecting different layer combinations" (e.g., only shallow + deep layers, or consecutive layers) on performance? If "scale embeddings" are removed, to what extent will the fusion effect of multi-scale features decrease?
2. Regarding the uniqueness of the innovation: What does the author consider to be the core difference between the "hierarchical multi-scale memory bank" and existing multi-layer feature fusion works (such as layer feature reuse in ViT-VQGAN)? Is there any experiment to prove the advantage of this memory bank in "simultaneously retaining macroscopic structures and microscopic details" (e.g., visualization of the contribution of features from different layers to reconstruction results)?
3. Regarding multimodal scalability: As a 1D Tokenizer, can MsTok be directly adaptable to image-text MLLMs (e.g., LLaVA)? If adaptation is required, is it necessary to modify the query mechanism or training strategy?

---

### Note · Authors · 2025-11-14

I have read and agree with the venue's withdrawal policy on behalf of myself and my co-authors.